# The Classification of Suspected Predominant Nociplastic Pain in People with Moderate and Severe Haemophilia: A Secondary Exploratory Study

**DOI:** 10.3390/biomedicines11092479

**Published:** 2023-09-07

**Authors:** Anthe Foubert, Nathalie Anne Roussel, Valérie-Anne Chantrain, Philip Maes, Lies Durnez, Sébastien Lobet, Catherine Lambert, Cédric Hermans, Mira Meeus

**Affiliations:** 1Research Group MOVANT, Department of Rehabilitation Sciences and Physiotherapy (REVAKI), University of Antwerp, 2000 Antwerp, Belgium; anthe.foubert@uantwerpen.be (A.F.); valerie-anne.chantrain@uantwerpen.be (V.-A.C.); lies.durnez@uantwerpen.be (L.D.); mira.meeus@uantwerpen.be (M.M.); 2Pain in Motion International Research Group, 1090 Brussels, Belgium; 3Faculté des Sciences de la Motricité, Université Catholique de Louvain, 1348 Louvain-La-Neuve, Belgium; 4Neuromusculoskeletal Lab (NMSK), Secteur des Sciences de la Santé, Institut de Recherche Expérimentale et Clinique, Université Catholique de Louvain, 1348 Louvain-la-Neuve, Belgium; sebastien.lobet@saintluc.uclouvain.be; 5Department of Paediatrics, University Hospital Antwerp, 2650 Edegem, Belgium; philip.maes@uza.be; 6Laboratory of Experimental Medicine and Pediatrics, University of Antwerp, 2000 Antwerp, Belgium; 7Haemostasis and Thrombosis Unit, Division of Hematology, Clinique Universitaires Saint-Luc, 1000 Brussels, Belgium; catherine.lambert@saintluc.uclouvain.be (C.L.); cedric.hermans@saintluc.uclouvain.be (C.H.); 8Secteur de Kinésithérapie, Cliniques Universitaires Saint-Luc, 1000 Brussels, Belgium; 9Department of Rehabilitation Sciences, Ghent University, 9000 Ghent, Belgium

**Keywords:** haemophilia, pain phenotyping, nociplastic pain, clinical criteria, classification, grading system

## Abstract

In people with haemophilia (PwH), joint pain is a major comorbidity that is often overlooked and under-treated. It is believed that, to ensure the most successful outcome, pain management should be tailored to the predominant pain phenotype (i.e., nociceptive, neuropathic and nociplastic). The 2021 clinical criteria and grading system for nociplastic pain, established by the International Association for the Study of Pain (IASP), emphasize the necessity of early-stage identification and predominant pain type classification. Consistent with findings in other chronic musculoskeletal pain conditions, studies suggest that a subgroup of PwH suffers from nociplastic pain, i.e., pain arising from altered nociception rather than structural damage, but this has not yet been explored in PwH. This study aimed to identify PwH with “unlikely”, “possible” and “probable” nociplastic pain and investigate differences in anthropometric, demographic and clinical characteristics and psychological factors between subgroups of PwH and healthy individuals.: The IASP clinical criteria and grading system were used to classify pain types in adult men with moderate or severe haemophilia recruited from two Belgian haemophilia treatment centres. Statistical analyses were applied to study between-subgroup differences. Of 94 PwH, 80 PwH (85%) were classified with “unlikely” and 14 (15%) with “at least possible” nociplastic pain (including 5 PwH (5%) with “possible” and 9 PwH (10%) with “probable” nociplastic pain). PwH in both the “unlikely” and “at least possible” nociplastic pain groups showed significantly higher levels of unhelpful psychological factors compared to healthy individuals. Additionally, age may partially account for the observed differences in body height and psychological factors. Larger sample sizes may be needed to detect more subtle between-group differences. study confirmed the presence of nociplastic pain in haemophilia, categorising a notable subgroup as individuals who experience at least possible nociplastic pain. These exploratory insights may provide a starting point for future studies and the development of more effective and tailored pain management.

## 1. Introduction

Pain is recognized as a significant concern in people with haemophilia (PwH), negatively impeding their daily activities and overall quality of life (QoL) [1,2]. This was emphasised by a German survey which showed that 86% of adults and 66% of children and adolescents experienced episodes of pain [3]. In addition to episodes of bleeding-related pain, PwH also experience pain associated with inflammation and multiarticular joint degeneration, with non-reversible end-stage haemophilic arthropathy [4]. Therefore, haemophilia can be considered as a chronic musculoskeletal disorder.

In contrast to chronic joint pain conditions such as osteoarthritis, the development of evidence-based guidelines for the management of pain in PwH faces many challenges. Indeed, the uncertainty about the presence of a bleed as a source of pain remains challenging both for patients and for health care professionals (HCPs) [5]. More importantly, HCPs’ limited understanding of pain, along with the succinct evaluation of pain during clinical consultations, creates obstacles to the development of a comprehensive biopsychosocial management strategy that takes into account biomedical, social and psychological factors contributing to pain and the underlying pain mechanisms [6]. The underassessment of pain and the lack of pain management options tailored to the underlying pain mechanism might explain the reduced QoL, as well as PwH’s low satisfaction rate with their pain treatment [3,7].

The importance of assessing the predominant pain mechanism in PwH was established in a review by our research group, and pain management should be adapted accordingly to ensure the most successful outcome [6]. Nowadays, it is assumed that pain in PwH is usually nociceptive in origin, due to the activation of nociceptors by an acute haemarthrosis or haemophilic arthropathy in a normally functioning somatosensory system. However, recent studies [8,9,10] revealed that a proportion of PwH demonstrated signs of neuropathic pain, i.e., pain caused by injury or lesion of the somatosensory system [11].

Moreover, in a variety of chronic musculoskeletal conditions, nociceptive and neuropathic pain mechanisms could not explain chronic pain in all patients. Subgroups of people with osteoarthritis [12], rheumatoid arthritis [13] and low back pain [14] demonstrated hypersensitivity in body regions with normal tissues and without signs of neuropathy (i.e., injury or lesion of the somatosensory system). For this phenomenon, the International Association for the Study of Pain (IASP) introduced the term nociplastic pain as a third underlying pain mechanism, in which pain results from altered somatosensory functioning without obvious activation of nociceptors or neuropathy (IASP Terminology (2017) https://www.iasp-pain.org/resources/terminology/, accessed on 2 April 2022).

Recent findings suggest that a subgroup of PwH is indeed more likely to suffer from nociplastic pain. The low associations between structural joint damage and pain experience [15,16], a rather widespread pain distribution and reduced pressure pain thresholds (PPTs) at several locations, including asymptomatic locations such as the sternum or forehead [10,17], suggest that these people present alterations in somatosensoric functioning, which may lead to nociplastic pain.

Diagnosing nociplastic pain is challenging since there is no gold standard methods. Consequently, patients experiencing such pain may feel that their symptoms are not taken seriously [18]. To address this issue, the IASP has developed clinical criteria and a grading system for nociplastic pain to help clinicians classify patients with predominant nociplastic pain [19]. With this grading system, it is possible to differentiate between “unlikely”, “possible” and “probable” nociplastic pain (Table 1). In contrast to the neuropathic pain grading system [20], the term “definite” nociplastic pain cannot be applied yet, due to the lack of validated diagnostic tests.

Members of the IASP Terminology Task Force strongly encourage field testing of the IASP grading system [19]. Therefore, the primary objective of this study was to determine the occurrence of nociplastic pain among a substantial sample of adult PwH by utilizing the IASP clinical criteria and grading system for nociplastic pain, thereby categorizing the participants into groups of unlikely, possible, and probable nociplastic pain. Our hypothesis was based on comparable chronic musculoskeletal conditions [18] assuming that there would be a subset of PwH exhibiting predominant nociplastic pain. The secondary aim was to analyse the dissimilarities in patient characteristics and psychological factors among four groups: PwH with unlikely, possible, and probable nociplastic pain and healthy individuals. These observations could potentially facilitate the development of more targeted treatment approaches specific to the identified pain phenotype.

## 2. Materials and Methods

The present study is reported in accordance with the Strengthening the Reporting of Observational Studies in Epidemiology (STROBE) guidelines [21].

### 2.1. Study Design and Setting

This field study is a secondary analysis of data obtained from observational studies aiming to gain insight into the complexity of pain in PwH [10,16,22]. The recruitment method has previously been described [10,16,22]. Briefly, data were collected between February 2020 and April 2022. PwH regularly followed at the Haemophilia Comprehensive Treatment Centre of the Cliniques universitaires Saint-Luc (Brussels, Belgium) and the Antwerp University Hospital (UZA) (Edegem, Belgium) underwent a comprehensive pain assessment after their consultation in the hospital setting. Healthy individuals were invited to the M^2^SENS lab (MOVANT, University of Antwerp, Belgium) to undergo identical tests. The ethical committee of the Antwerp University Hospital approved the multicentre study (B300201942304) and all participants provided written informed consent before study participation.

### 2.2. Participants

To be included, PwH had to be adult men over 18 years old with moderate (FVIII or FIX activity between 2 and 5 IU/dL) or severe (FVIII or FIX activity < 1 IU/dL) haemophilia A or B and provide evidence that their haemophilia treatment regimen is stable (i.e., an unmodified treatment during the last six months, verified by the existing patients’ logbooks). PwH suffering from known neuropathies with definite medical causes independent from the haemophilia (e.g., diabetic polyneuropathy) were excluded as this might influence pain assessment [23]. For the same reason, PwH with acute pain due to a joint bleed occurring in the month preceding study participation were excluded as well. In case of doubt, point of care ultrasound was used to check for the presence of blood in the joint. Additional exclusion criteria for this field study were: 1. PwH without pain and 2. PwH who had not completed the questionnaires.

Exclusion criteria for the pain-free individuals were: 1. known pain diseases or conditions influencing nociceptive processing (e.g., rheumatologic, inflammatory, metabolic, malignant diseases), 2. having any pain/discomfort with a score of >0 on the 10-point numeric rating scale (NRS) at the time of assessment [24] and 3. diagnosis of depression or other psychiatric complaints.

### 2.3. The IASP Clinical Criteria and Grading System for Nociplastic Pain Applied to Pain in PwH

The step-by-step clinical reasoning process below describes how the authors applied the IASP clinical criteria and grading system for nociplastic pain to PwH suffering from chronic pain [19]. For each step, the assessment tools or questionnaires used to evaluate whether PwH met the criterion are explained. In addition, the cut-offs available in the literature that supported the authors’ decision are stipulated. Table 1 summarises the clinical reasoning process.


**Step 1—A chronic pain duration**


Based on the inclusion criteria, no PwH without pain or with acute pain were included.


**Step 2—A regional/multifocal/widespread pain distribution**


The Brief Pain Inventory (BPI) body chart pain drawing was used to determine whether PwH had a regional/multifocal/widespread rather than a discrete pain distribution. To objectify the assessment, individuals who indicated four or more painful sites on the body chart were considered as having regional/multifocal/widespread pain [25].


**Step 3—The pain cannot entirely be explained by nociceptive mechanisms**


PwH in which nociceptive mechanisms are considered to be entirely responsible for their pain should be classified as PwH with nociceptive pain. However, the decision on this criterion is difficult and depends on the clinical judgment of the investigator; this is recognized by the task force as a limitation [19] since the presence of “nociceptive” pain does not exclude the possibility of concurrent nociplastic pain (i.e., mixed pain) when the pain distribution exceeds the identifiable source of nociception (i.e., widespread pain) [19]. Therefore, we assumed that it would not be possible to reliably identify nociceptive pain as the main driver of pain in PwH with a widespread pain distribution and they moved on to step 4.


**Step 4—The pain cannot entirely be explained by neuropathic mechanisms**


Individuals with known neuropathies with definite medical causes independent from the haemophilia were excluded prior to study participation. However, to detect PwH with undiagnosed neuropathic pain, we followed the guideline for the classification of neuropathic pain [20]. Accordingly, PwH with a score of ≥4/10 on the DN4 questionnaire (i.e., indicating signs of neuropathic pain) and a neuroanatomically plausible pain distribution were considered as having possible neuropathic pain for which further confirmatory testing was recommended [20,26]. Similar to nociceptive pain, the presence of neuropathic pain does not exclude the possibility of concurrent nociplastic pain (i.e., mixed pain) [19]. Therefore, in PwH with only a positive DN4 score without a plausible pain distribution, the neuropathic pain cannot be considered as entirely responsible for the pain and they move on to step 5.


**Step 5—Evoked hypersensitivity phenomena**


The presence of evoked pain hypersensitivity was evaluated according to the previous results of three QST outcomes: mechanical pressure pain thresholds (PPT) and cold and heat pain thresholds (CPT and HPT). PPT were evaluated using a digital algometer (Wagner Instruments©, Greenwich, CT, USA) at the medial joint space of the knees and anterior aspect of the talocrural joint line of the ankles [10]. For each PwH, the mean value of two PPT assessments performed at a self-reported painful joint without a prosthesis was used for comparison with pain-free individuals. Hypersensitivity to mechanical pressure pain was considered when the PPT value exceeded a Z-score of 1.96 compared to healthy individuals in ≥50% of the painful joints [27,28].

CPT and HPT were determined using a 30 × 30 mm thermode of a TSA-2 device (Medoc©, Ramati-Yishai, Israel) attached at the dominant wrist [29,30]. Starting at 32 °C, the temperature increased or decreased by 1 °C per second. Maximums were set at 0 °C and 50 °C to prevent tissue damage. Participants had to press a button as soon as the temperature became painful. For both parameters, the average of three consecutive recordings was used to determine the threshold [27]. Hypersensitivity to cold or heat was considered when the threshold exceeded a Z-score of 1.96 compared to healthy individuals [27,28]. PwH presenting at least one of the three evoked pain hypersensitivity phenomena were considered to fulfil step 5. Consequently, PwH who met all five steps of the grading system were classified as having “possible” nociplastic pain, while the others were classified as having “unlikely” nociplastic pain.


**Step 6–A history of pain hypersensitivity**


A thorough patient interview may suffice to examine a history of pain hypersensitivity to touch, pressure, movement or cold and heat [19]. However, we had the possibility of using the QST results from step 5 to determine whether PwH had a history of pain hypersensitivity [19]. Consequently, PwH who fulfilled step 5 also fulfilled step 6.


**Step 7—The presence of comorbidities**


To evaluate the final step, we followed the additional recommendations of Nijs et al. [31] and used the results from the Central Sensitization Inventory (CSI) questionnaire (part A), since it covers the listed comorbidities such as increased sensitivity to sound, light and/or odours, sleep disturbance with frequent nocturnal awakenings, fatigue and cognitive problems. Comorbidities are scored using a numerical rating scale: never (0), rarely (1), sometimes (2), often (3), and always (4) present [32]. Since no recommendations are yet available on what score the item should have to be defined as a comorbidity, we chose a strict cut-off. PwH achieving at least a score of 3 (often present) for “two or more” comorbidities were considered to meet this criterion and, therefore, were classified as having “probable” nociplastic pain. Indeed, this contrasts with having “one” comorbidity in the IASP clinical criteria for nociplastic pain, but as a recent study highlighted, thorough clinical reasoning is needed to decide whether comorbidities can contribute to pain phenotyping [33]. For example, sleep disturbances are commonly reported [34] in PwH, including in people with mild haemophilia [35] and in the general population [36]. Besides part A of the CSI questionnaire, part B was used to collect additional information on specific central sensitivity syndromes that were diagnosed in the past.

### 2.4. Comparison between Groups

After the application of the grading system, PwH were divided into three subgroups: “unlikely” “possible” and “probable” nociplastic pain. Additionally, patient characteristics (i.e., anthropometric, demographic and clinical details) and psychological factors were compared to investigate intergroup differences. Psychological factors were investigated using a battery of validated self-reported questionnaires. The Pain Catastrophizing Scale (PCS) examined the degree of catastrophic thinking in the content domains of rumination, magnification and helplessness. Higher scores indicated higher levels of pain catastrophizing [37]. The Hospital Anxiety and Depression Scale (HADS) was used to investigate symptoms of anxiety and depression. Higher scores indicated more symptoms of anxiety and depression [38]. Fear-avoidance beliefs were measured using the Fear Avoidance and Beliefs Questionnaire (FABQ). Higher scores indicated elevated fear-avoidance beliefs. Health-related quality of life (HR-QoL) was evaluated using the EuroQol-5 Dimensions 5 Levels questionnaire (EQ-5D-5L). A health utility index (EQ-HUI) score was calculated to rate the impact of their disease on mobility, self-care, usual activities, pain/discomfort and anxiety/depression. Using the visual analogue scale (EQ-VAS), participants were asked to rate their health state from 0 “worst imaginable” to 100 “best imaginable” [39].

### 2.5. Statistical Analyses

Statistical data analyses were carried out using the IBM Statistical Package for Social Sciences Version 28 (SPSS, IBM Corporation, Armonk, NY, USA). Descriptive data for continuous variables are presented as means and standard deviations (±SD); data for categorical variables are presented as percentages. The Student’s *t*-test was applied to compare PwH and healthy individuals. One-way analysis of variance (ANOVA) was used to compare patient characteristics and psychological factors between the subgroups formed after applying the IASP clinical criteria and grading system of nociplastic pain. Additionally, analysis of covariance (ANCOVA) was used to compare patient characteristics and psychological factors while controlling for age. Post hoc analyses following ANOVA and ANCOVA were adjusted for multiple hypothesis testing using a Bonferroni correction (α = 0.05/3 = 0.017). Frequency differences for categorical variables were tested using the Chi-squared test. Significance level was set at 0.05.

## 3. Results

### 3.1. Participants

Clinical data from 94 PwH and 41 pain-free healthy individuals, who underwent a comprehensive pain assessment between February 2020 and April 2022, were available to apply to the grading system. Demographic, anthropometric and clinical characteristics are presented in Table 2.

### 3.2. The Classification of Nociplastic Pain

For each step of the grading system, the number and percentage of PwH that fulfilled the criteria are presented below. Figure 1 provides a visual overview of the clinical reasoning process.


**Step 1—A chronic pain duration**


In total, 94 PwH (100%) suffering from chronic pain were included.


**Step 2—A regional/multifocal/widespread pain distribution**


According to the BPI body chart, 35 PwH (37%) indicated at least four painful body sites and were therefore considered as having regional/multifocal/widespread pain. The remaining 59 PwH (63%) indicated less than four painful sites and were therefore removed from the grading system, as they were unlikely to have nociplastic pain.


**Step 3—The pain cannot entirely be explained by nociceptive mechanisms**


As mentioned in the Methods section, we did not exclude participants based on this condition, so the remaining 35 PwH (37%) were considered to move on to the next step.


**Step 4—The pain cannot entirely be explained by neuropathic mechanisms**


Of the 35 PwH, 16 scored ≥ 4/10 on the DN4, but none of them had a positive DN4 score together with a neuroanatomically plausible pain distribution on the BPI body chart. For this reason, the presence of pain entirely explained by neuropathic pain was unlikely, and all 35 PwH (37%) moved further to the next steps.


**Step 5—Evoked hypersensitivity phenomena**


Table 3 presents the QST results of the 35 PwH. Fourteen of them (40%) had a cold, heat or pressure pain threshold exceeding the Z-score of 1.96 compared to healthy individuals, indicating clinical signs of evoked pain hypersensitivity. Consequently, these 14 PwH met the requirements of the first five steps and could therefore be classified as PwH having “possible” nociplastic pain.


**Step 6—A history of pain hypersensitivity**


As mentioned in the Methods section, all 14 PwH who fulfilled step 5 also fulfilled step 6.


**Step 7—The presence of comorbidities**


Of the 14 remaining PwH, nine (10%) had at least a score of 3 (‘often present’) for two or more comorbidities of the CSI (part A). Therefore, these nine PwH (10%) could be classified as having “probable” nociplastic pain. Among those, four PwH (44%) reported having been diagnosed with a central sensitivity syndrome in the past.

### 3.3. Comparison between Groups

According to the IASP clinical criteria for nociplastic pain, PwH were classified into three subgroups: PwH with “unlikely” nociplastic pain (n = 80), PwH with “possible” nociplastic pain (n = 5), and PwH with “probable” nociplastic pain (n = 9). Given the small sample size, the “possible” and “probable” nociplastic pain subgroups were merged into one “at least possible” nociplastic pain subgroup (n = 14) for analyses of differences between the PwH subgroups and the group of healthy individuals (n = 41) in which three groups were compared (Table 4).

One-way ANOVA showed significant group differences in height (*p* = 0.004), body mass index (BMI, *p* = 0.038), use of self-reported pain medication (*p* < 0.001) and psychological factors (all *p* < 0.012). Pairwise post hoc comparisons (Bonferroni-corrected) revealed that PwH with “at least possible” nociplastic pain were significantly shorter compared to healthy individuals (*p* = 0.005) but could not demonstrate significant between-group differences for BMI. For the psychological factors, both PwH with “unlikely” nociplastic pain and PwH with “at least possible” nociplastic pain showed significantly higher mean scores compared to healthy individuals (all *p* < 0.012). There were no significant differences between PwH with “unlikely” and “at least possible” nociplastic pain.

When taking age into account, ANCOVA revealed significant group differences for body height (*p* = 0.013) and psychological factors (all *p* < 0.013). Pairwise post hoc comparison revealed that both PwH with “unlikely” and “at least possible” nociplastic pain showed significantly higher mean scores for psychological factors compared to healthy individuals (all *p* < 0.016). There were no significant differences between PwH with “unlikely” and “at least possible” nociplastic pain.

## 4. Discussion

Application of the IASP clinical criteria and grading system of nociplastic pain revealed that 80/94 PwH (85%) could be classified with “unlikely” nociplastic pain, 5/94 PwH (5%) with “possible” and 9/94 PwH (10%) with “probable” nociplastic pain. When merging the latter two into a group with “at least possible” nociplastic pain, these 14 PwH (15%) were significantly shorter than healthy individuals. Both PwH with “at least possible” and “unlikely” nociplastic pain showed significantly higher levels of unhelpful psychological factors compared to healthy individuals. When controlling for age, again, both PwH groups (i.e., “unlikely” and “at least possible” nociplastic pain) showed significantly higher levels of unhelpful psychological factors compared to healthy individuals. These findings suggest that age may partially account for the observed group differences in body height and psychological factors.

Indeed, considering that many conditions clinically show a mixed presentation of pain mechanisms [40], we did not intend to utilize the criteria to identify a single pain mechanism but rather to investigate the predominant pain mechanism in PwH. Since this is the first field study applying the clinical criteria for nociplastic pain in a haemophilia population, a cautious interpretation is needed. Therefore, we would like to highlight the challenges encountered during the course of this study.

### 4.1. A Clear Definition of Regional/Multifocal/Widespread Pain Is Needed

Pain drawings have recently been recommended as reliable and valid tools to evaluate an individual’s pain distribution [31]. However, this recommendation does not provide a clear definition or cut-off when a patient presents a regional, multifocal or widespread pain distribution which is indicative of nociplastic pain [19]. Consistent with a previous study, we considered individuals who indicated four or more painful body sites as having regional pain [25]. However, a clear definition is needed when this criterion is used for identifying nociplastic pain, to avoid the decision depending entirely on the expertise and clinical reasoning of HCPs.

### 4.2. Clinical Criteria or a Grading System for Nociceptive Pain Is Needed

Currently, no single guideline or criteria exist for nociceptive pain. Therefore, the IASP Terminology Task Force recognises the reliance on clinical judgement to decide whether nociceptive pain mechanisms can be considered entirely responsible for the person’s pain as a major limitation [19]. For this reason, we assumed that it would not be possible to reliably identify nociceptive pain as the main driver of pain in PwH. Thus, further research is needed to elaborate this.

### 4.3. The Evaluation of Evoked Pain Hypersensitivity Needs Clarification

Again, no clear definition or cut-off values are provided to define evoked pain hypersensitivity as a clinical criterion for nociplastic pain [19]. Therefore, by analogy with the field study of Schmidt et al. (2022) [28,41], we defined hypersensitivity as a QST threshold exceeding a Z-score of 1.96 compared to healthy individuals. In addition, information about the exact body location where the QST analysis should be performed is lacking [19]. Since we are convinced that QST in one painful body region might overestimate the presence of pain hypersensitivity, we opted for a stricter cut-off, namely that hypersensitivity had to be present in ≥50% of the painful body regions. QST at remote locations, such as the dominant wrist in our case, seems more reliable because it immediately investigates secondary hyperalgesia [42]. Moreover, we support the suggestion of Schmidt et al. (2022) to switch the sequence of step 4 and step 5, as we agree that clinical pain assessment should follow the patient’s history and self-reported questionnaires by analogy with the grading system for neuropathic pain [20]. Since QST is not mandatory to assess pain hypersensitivity [19], it would make more sense for clinicians who cannot include QST in daily practice.

### 4.4. The Impact of Assessing Comorbidities

Our subgroup analysis revealed that PwH with “at least possible” nociplastic pain were significantly shorter and reported higher levels of unhelpful psychological factors (i.e., pain catastrophizing and anxiety) compared to healthy individuals. Although these differences may be related to the higher age of PwH with “at least possible” nociplastic pain, we should continue to investigate comorbidities (e.g., through the CSI) in PwH. Since studies show that obesity [43], sleep disturbances [44] and unhelpful psychological factors [45] are risk factors for the development of chronic pain, we should include these risk factors in the biopsychosocial pain management approach.

### 4.5. Strengths and Limitations

As mentioned above, this is the first field test application of the IASP clinical criteria for nociplastic pain in a large sample of PwH, which may motivate other researchers investigating chronic musculoskeletal pain conditions (including haemophilia) to explore it further. Secondly, the task force indicated the dependence on clinical judgement of the investigator as a major limitation of the clinical criteria [19]. To overcome this limitation, we assessed the criteria as objectively as possible based on validated QST protocols, existing literature and cut-offs [27]. A limitation of this study is that, for the subgroup analysis, we had to combine the subgroups having “possible” and “probable” nociplastic pain into one group with “at least possible” nociplastic pain. A power analysis showed that comparing a group with 14 observations to groups with 80 and 41 observations offers 80% power to detect an effect size (Cohen’s D) of 0.79 and 0.86 standard deviations at a significance level of 0.05. Since this is the first study ever investigating this, no reference data are available to judge whether these differences are realistic. With a larger sample size, more subtle differences between the “at least possible” group and the two other groups can be detected.

### 4.6. Clinical Implications and Implications for Future Research

The IASP task force has strongly encouraged field tests of the clinical criteria, but, to date, clinimetric and psychometric properties have not been investigated. Therefore, validation studies are urgently needed to ensure that future field tests follow validated and reliable procedures. In addition, the present study provides information for future experimental studies, for example, studies investigating pain management strategies tailored to a predominant nociplastic pain mechanism (i.e., interventional studies focussing on psychological treatment modalities [45] or trials of centrally acting pain medications such as antidepressants or serotonin-norepinephrine reuptake inhibitors) [6,46].

## 5. Conclusions

A subgroup of PwH could be classified as having at least possible nociplastic pain. This early identification of PwH with predominant nociplastic pain might be an important step towards more effective and tailored pain management. Since this field study applied the IASP clinical criteria for nociplastic pain in haemophilia for the first time, reference data are not yet available and further studies with larger sample sizes may be needed to detect more subtle differences between groups. Moreover, further studies examining the clinimetric and psychometric properties of the IASP clinical criteria are needed as we believe that sound criteria could help HCPs in steering their pain approach.

## Figures and Tables

**Figure 1 biomedicines-11-02479-f001:**
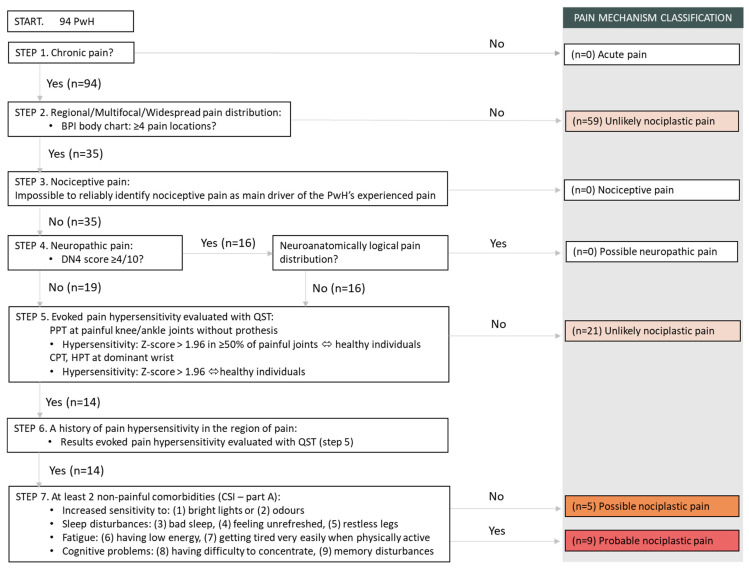
Clinical reasoning process on the application of the IASP clinical criteria and grading system for nociplastic pain in PwH.

**Table 1 biomedicines-11-02479-t001:** The IASP clinical criteria and grading system for nociplastic pain applied to pain in PwH.

**STEP 1.**	**The pain is chronic:** PwH with chronic pain will fulfil this step.
**STEP 2.**	**The pain has a regional/multifocal/widespread distribution:** ≥4 painful body sites on the BPI-Body chart.
**STEP 3.**	**The pain cannot entirely be explained by nociceptive mechanisms:** All PwH will fulfil this step, since it is impossible to reliably identify nociceptive pain as the main driver of the PwH’s experienced pain.
**STEP 4.**	**The pain cannot entirely be explained by neuropathic mechanisms:** PwH without possible neuropathic pain will fulfil this step. (Possible neuropathic pain: a DN4 score of ≥4/10 and a neuroanatomically plausible pain distribution).
**STEP 5.**	**Evoked hypersensitivity phenomena:** PwH presenting evoked hypersensitivity evaluated with QST will fulfil this step: - Pressure Pain Threshold at painful knee/ankle joints without prothesis: Hypersensitivity: Z-score > 1.96 in ≥50 of painful joints ⇔ healthy individuals - Cold & Heat Pain Threshold at dominant wrist: Hypersensitivity: Z-score > 1.96 ⇔ healthy individuals
Possible nociplastic pain: PwH who fulfil all 5 steps. Unlikely nociplastic pain: PwH who fulfil none or some of the steps.	
**STEP 6.**	**A history of pain hypersensitivity:** When QST results are present they can be used to determine whether PwH have a history of pain hypersensitivity. PwH who present pain hypersensitivity in step 5 will automatically fulfil step 6.
**STEP 7.**	**The presence of comorbidities:** PwH will fulfil step 7 if they achieve at least a score of 3 (often present) for ≥2 comorbidities on the CSI part A: - Increased sensitivity to: (1) bright lights or (2) odours - Sleep disturbances: (3) bad sleep, (4) feeling unrefreshed, (5) restless legs - Fatigue: (6) having low energy, (7) getting tired very easily when physically active - Cognitive problems: (8) having difficulty to concentrate, (9) memory disturbances
Probable nociplastic pain: PwH who fulfil all 7 steps.	

Abbreviations: PwH, People with Haemophilia; BPI, Brief Pain Inventory; DN4, Douleur Neuropathique en 4 questions; QST, Quantitative Sensory Testing; CSI, Central Sensitization Inventory.

**Table 2 biomedicines-11-02479-t002:** Demographic, anthropometric and clinical characteristics of the participants.

	PwH (n = 94)		Healthy Individuals (n = 41)		*p*-Value ^a^
	Mean ± SD (Range)	n (%)	Mean ± SD (Range)	n (%)	
Age (years)	41.7 ± 16.9 (18–81)	-	38.8 ± 17.2 (18–79)	-	0.372
Weight (kg)	80.8 ± 16.1 (48.7–128)	-	77.5 ± 10.9 (60–104)	-	0.177
Height (m)	1.77 ± 0.06 (1.60–1.88)	-	1.80 ± 0.07 (1.64–1.93)	-	0.005
BMI (kg/m^2^)	26.0 ± 4.9 (16.9–40.9)	-	24.1 ± 3.3 (18.6–31.1)	-	0.008
Type of haemophilia—severity					-
A—severe	-	62 (66.0%)	-	-	
A—moderate		12 (12.8%)			
B—severe		11 (11.7%)			
B—moderate	-	9 (9.6%)	-	-	
Treatment regimen					-
On-demand	-	11 (11.7%)	-	-	
Prophylaxis	-	83 (88.3%)	-	-	
Self-reported use of pain medication		24 (25.5%)		0 (0%)	<0.001 ^b^
Non-opioid analgesics	-	14 (14.9%)	-	-	
Non-opioid + weak opioid analgesics	-	1 (1.1%)	-	-	
Non-opioid analgesics + recombinant factor	-	9 (9.6%)	-	-	
Positive HIV	-	6 (6.4%)	-	-	-
Hepatitis C					-
Negative	-	44 (46.8%)	-	-	
Succesfully treated for HCV (negative viral load)	-	50 (53.2%)	-	-	

Data are presented as mean ± SD for continuous variables and as frequency counts (%) for categorical variables. Abbreviations: PwH, people with haemophilia; SD, standard deviation; kg, kilograms; m, meters; BMI, body mass index; HIV, human immunodeficiency virus; HCV, hepatitis C virus. ^a^ Student’s *t*-test, *p* < 0.05. ^b^ Chi-squared tests, *p* < 0.05.

**Table 3 biomedicines-11-02479-t003:** Results Quantitative Sensory Testing.

QST	Healthy Individuals (n = 41)		PwH (n = 35)			
	n	Mean ± SD	n	Mean ± SD	n(%) Z-Score > 1.96	n(%) Z-Score PPT > 1.96 in ≥50% of Painful Joints or Z-Score CPT/HPT > 1.96
CPT wrist (°C)	41	6.67 ± 7.98	33 ^a^	10.19 ± 9.77	7 (20%)	14 (40%)
HPT wrist (°C)	41	46.36 ± 2.60	33 ^a^	45.90 ± 3.22	3 (9%)	
PPT knee left (N)	41	67.04 ± 20.18	14 ^b^	36.72 ± 16.72	6 (17%)	
PPT knee right (N)	41	68.64 ± 22.37	10 ^b^	43.73 ± 16.25	1 (3%)	
PPT ankle left (N)	41	68.94 ± 19.84	22 ^b^	41.25 ± 17.14	7 (20%)	
PPT ankle right (N)	41	68.87 ± 22.59	24 ^b^	43.42 ± 19.77	3 (9%)	

^a^ Two missing values for this outcome parameter. ^b^ Number of PwH reporting the joint as painful (but without prothesis). Abbreviations: QST, Quantitative Sensory Testing; CPT, Cold Pain Threshold; HPT, Heat Pain Threshold; PPT, Pressure Pain Threshold; N, Newton.

**Table 4 biomedicines-11-02479-t004:** Subgroup analysis by ANOVA and ANCOVA (adjusted for age).

	PwH with “Unlikely” Nociplastic Pain (n = 80)	PwH with “Possible” Nociplastic Pain (n = 14)	Healthy Individuals (n = 41)	ANOVA *p*-Value	*p*-Value * Pairwise Comparison Subgroups (ANOVA)	ANCOVA *p*-Value	*p*-Value * Pairwise Comparison Subgroups (ANCOVA)
	Mean ± SD (Range) or n (%)	Mean ± SD (Range) or n (%)	Mean ± SD (Range) or n (%)				
Age (years)	40.3 ± 15.9 (18–74)	49.5 ± 21 (19–81)	38.8 ± 17.2 (18–79)	0.117	-	-	-
Weight (kg)	80.6 ± 15.1 (48.7–128)	82 ± 21.6 (50–117)	77.5 ± 10.9 (60–104)	0.480	-	0.526	-
Height (m)	1.77 ± 0.06 (1.62–1.88)	1.73 ± 0.08 (1.60–1.87)	1.80 ± 0.07 (1.64–1.93)	0.004	0.005 (Possible vs. Healthy)	0.013	-
BMI (kg/m^2^)	25.8 ± 4.6 (16.9–40.9)	27.3 ± 6.7 (18.6–37.9)	24.1 ± 3.3 (18.6–31.1)	0.038	-	0.069	-
Type of haemophilia—severity				0.102 ^a^	-	-	-
A—severe	53 (66%)	9 (64%)	-				
A—moderate	12 (15%)	-	-				
B—severe	7 (9%)	4 (29%)	-				
B—moderate	8 (10%)	1 (7%)	-				
Treatment regimen				0.412 ^a^	-	-	-
On-demand	11 (14%)	-	-				
Prophylaxis	56 (70%)	11 (79%)	-				
Emicizumab	13 (16%)	3 (21%)	-				
Gene therapy		-	-				
Self-reported use of pain medication	20 (25%)	4 (28%)	0 (0%)	<0.001 ^a^	-	-	-
Non-opioid analgesics	12 (15%)	2 (14%)	-				
Non-opioid + weak opioid analgesics	1 (1%)	-	-				
Non-opioid + strong opioid analgesics	-	-	-				
Non-opioid analgesics + recombinant factor	7 (9%)	2 (14%)	-				
HADS							
Anxiety (max. 21)	6.0 ± 3.8 (0–18)	6.3 ± 3.0 (1–10)	3.6 ± 2.7 (0–12)	<0.001	<0.001 (Unlikely vs. Healthy)	<0.001	<0.001 (Unlikely vs. Healthy)
							0.016 (Possible vs. Healthy)
Depression (max. 21)	4.3 ± 3.5 (0–15)	5.7 ± 2.6 (0–9)	2.2 ± 1.9 (0–7)	<0.001	<0.001 (Unlikely vs. Healthy)	<0.001	<0.001 (Unlikely vs. Healthy)
					<0.001 (Possible vs. Healthy)		<0.001 (Possible vs. Healthy)
PCS							
Total (max. 52)	14.2 ± 11.1 (0–47)	20.3 ± 12.6 (0–37)	7.2 ± 7.1 (0–23)	<0.001	0.002 (Unlikely vs. Healthy)	<0.001	0.002 (Unlikely vs. Healthy)
					<0.001 (Possible vs. Healthy)		<0.001 (Possible vs. Healthy)
PCS Rumination (max. 16)	4.9 ± 4.4 (0–16)	7.2 ± 4.5 (0–13)	3.4 ± 3.7 (0–14)	0.012	0.012 (Possible vs. Healthy)	0.013	0.013 (Possible vs. Healthy)
PCS Magnification (max. 12)	3.0 ± 2.4 (0–10)	4.6 ± 3.3 (0–9)	1.3 ± 1.6 (0–6)	<0.001	<0.001 (Unlikely vs. Healthy)	<0.001	<0.001 (Unlikely vs. Healthy)
					<0.001 (Possible vs. Healthy)		<0.001 (Possible vs. Healthy)
PCS Helplessness (max. 24)	6.3 ± 5.4 (0–23)	8.4 ± 6.4 (0–21)	2.5 ± 2.6 (0–10)	<0.001	<0.001 (Unlikely vs. Healthy)	<0.001	<0.001 (Unlikely vs. Healthy)
					<0.001 (Possible vs. Healthy)		<0.001 (Possible vs. Healthy)
FABQ							
Physical activity (max. 24)	13.8 ± 6.2 (0–24)	16.1 ± 6.4 (7–24)	9.2 ± 7.5 (0–24)	<0.001	0.001 (Unlikely vs. Healthy)	<0.001	0.001 (Unlikely vs. Healthy)
					0.003 (Possible vs. Healthy)		0.006 (Possible vs. Healthy)
EQ-5D-5L							
EQ-HUI (max. 1)	0.7 ± 0.2 (0–1)	0.6 ± 0.2 (0.2–0.9)	1.0 ± 0.1 (0.7–1)	<0.001	<0.001 (Unlikely vs. Healthy)	<0.001	<0.001 (Unlikely vs. Healthy)
					<0.001 (Possible vs. Healthy)		<0.001 (Possible vs. Healthy)
EQ-VAS (max. 100)	70.4 ± 15.3 (27–80)	69.0 ± 19.8 (30–100)	84.9 ± 8.5 (67–100)	<0.001	<0.001 (Unlikely vs. Healthy)	<0.001	<0.001 (Unlikely vs. Healthy)
					0.001 (Possible vs. Healthy)		<0.001 (Possible vs. Healthy)

Data are presented as mean ± SD for continuous variables and as frequency counts (%) for categorical variables. Abbreviations: PwH, people with haemophilia; SD, standard deviation; kg, kilograms; m, meters; BMI, body mass index; HIV, human immunodeficiency virus; HCV, hepatitis C virus; HADS, hospital anxiety and depression scale; PCS, pain catastrophizing scale; FABQ, fear-avoidance and beliefs questionnaire; EQ-5D-5L, EuroQol 5 dimensions 5 levels questionnaire; EQ-HUI, EuroQol health utility index; EQ-VAS, EuroQol visual analogue scale. ^a^ *p*-values calculated using Chi-squared tests. * *p*-values for posthoc analysis corrected with Bonferroni method (α = 0.05/3 = 0.017).

## Data Availability

Data available on request from the authors.

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
