# Peer review of "The Classification of Suspected Predominant Nociplastic Pain in People with Moderate and Severe Haemophilia: A Secondary Exploratory Study"

_biomedicines, 2023, doi:10.3390/biomedicines11092479_

Round 1

Reviewer 1 Report

The authors of the manuscript focused on  identify in hemophilic patients PwH  nociplastic pain and investigate differences in anthropometric, demographic, clinical characteristics and psychological factors between patients and healthy control. Joint pain is common in haemophilia and may be acute or chronic. Effective pain management in haemophilia is essential to reduce the burden that pain imposes on patients. However, the choice of appropriate pain-relieving measures is challenging, as there is a complex interplay of factors affecting pain perception. This can manifest as differences in patients’ experiences and response to pain, which require an individualized approach to pain management. The manuscript is well structured, but some parts are missing some important facts that authors should add.

Page 1: ....In addition to episodes of bleeding-related pain,

PwH also experience pain associated with inflammation and joint degeneration, with non-reversible end-stage haemophilic arthropathy. Therefore, haemophilia can be considered as a chronic musculoskeletal disorder. This citation is missing: Hemophilic patients present with multiarticular joint degeneration (hemophilic arthropathy). Authors should cite the most recent reference stating this: ,, Perioperative Monitoring with Rotational Thromboelastometry in a Severe Hemophilia A Patient Undergoing Elective Ankle Surgery. Semin Thromb Hemost. 2023 Apr 19. doi: 10.1055/s-0043-57009“

Figure and table in the text are very clearly written.

I have to say that with these 49 references there half of the references are from the last 5 years. Authors should cite more recent references

Author Response

Dear reviewer, 

Please find a point-by-point response in the attachment. 

Reviewer 2 Report

The authors investigate differences in anthropometric, demographic, clinical characteristics and psychological factors between subgroups of PwH and healthy individuals.

Please include more information in the Abstract to introduce your content.

I suggest the authors make a concise conclusion in the contents.

The manuscript requires careful editing.

Author Response

(The authors gave the same response as above.)

Round 2

Reviewer 1 Report

The presented manuscript has been corrected in response to the suggestions. The authors have followed the recommendations of the reviewer. After the revision, the provided data and addition of the results became more clear. I would like to thank the authors for resubmitting the manuscript and explaining the obscure points from the previous version.